# Higher-Order Interactions and Their Duals Reveal Synergy and Logical Dependence beyond Shannon-Information

**DOI:** 10.3390/e25040648

**Published:** 2023-04-12

**Authors:** Abel Jansma

**Affiliations:** 1MRC Human Genetics Unit, Institute of Genetics & Cancer, University of Edinburgh, Edinburgh EH8 9YL, UK; a.a.a.jansma@sms.ed.ac.uk; 2Higgs Centre for Theoretical Physics, School of Physics & Astronomy, University of Edinburgh, Edinburgh EH8 9YL, UK; 3Biomedical AI Lab, School of Informatics, University of Edinburgh, Edinburgh EH8 9YL, UK

**Keywords:** higher-order, information, entropy, synergy, triadic, Möbius inversions, Ising model, lattices

## Abstract

Information-theoretic quantities reveal dependencies among variables in the structure of joint, marginal, and conditional entropies while leaving certain fundamentally different systems indistinguishable. Furthermore, there is no consensus on the correct higher-order generalisation of mutual information (MI). In this manuscript, we show that a recently proposed model-free definition of higher-order interactions among binary variables (MFIs), such as mutual information, is a Möbius inversion on a Boolean algebra, except of surprisal instead of entropy. This provides an information-theoretic interpretation to the MFIs, and by extension to Ising interactions. We study the objects dual to mutual information and the MFIs on the order-reversed lattices. We find that dual MI is related to the previously studied differential mutual information, while dual interactions are interactions with respect to a different background state. Unlike (dual) mutual information, interactions and their duals uniquely identify all six 2-input logic gates, the dy- and triadic distributions, and different causal dynamics that are identical in terms of their Shannon information content.

## 1. Introduction

### 1.1. Higher-Order Interactions

All non-trivial structures in data or probability distributions correspond to dependencies among the different features, or variables. These dependencies can be present among pairs of variables, i.e., pairwise, or can be *higher-order*. A dependency, or interaction, is called higher-order if it is inherently a property of more than two variables and if it cannot be decomposed into pairwise quantities. The term has been used more generally to refer simply to complex interactions, as for example in [1] to refer to changes in gene co-expression over time; in this article, however, it is used only in the stricter sense defined in Section 2.

The reason such higher-order structures are interesting is twofold. First, higher-order dependence corresponds to a fundamentally different kind of communication and interaction among the components of a system. If a system contains higher-order interactions, then its dependency structure cannot be represented by a graph and requires a hypergraph, where a single ‘hyperedge’ can connect more than two nodes. It is desirable to be able to detect and describe such systems accurately, which requires a good understanding of higher-order interactions. Second, higher-order interactions might play an important role in nature, and have been identified in various interaction networks, including genetic [2,3,4,5], neuronal [6,7,8,9,10], ecological [11,12,13], drug interaction [14], social [15,16,17], and physical [18,19] networks. Furthermore, there is evidence that higher-order interactions are responsible for the rich dynamics [20] or bistability [21] in biological networks; for example, synthetic lethality experiments have shown that the trigenic interactions in yeast form a larger network than the pairwise interactions [4].

Despite this, purely pairwise descriptions of nature have been remarkably successful, which the authors of [22,23] attribute to the fact that there are regimes in terms of the strength and density of coupling among the variables within which pairwise descriptions are sufficient. Alternatively, it may be attributed to the fact that higher-order interactions have been understudied and their effects underestimated. Currently, perhaps the most promising method of quantifying higher-order interactions is information theory. The two most commonly used quantities are mutual information and its higher-order generalisation (used in, e.g., [24,25]) and the total correlation (introduced in [26] and recently used in [27]). However, one particular problem of interest that total correlation and mutual information do not address is that of synergy and redundancy. Given a set of variables with an *n*th-order dependency, what part of that is exclusively *n*th-order (called the synergistic part), and what part can be found in a subset of m<n variables as well (the redundant part)? Quantifying the exact extent to which shared information is synergistic is an open problem, and is most commonly addressed using *partial information decomposition* [28], which has been applied mainly in the context of theoretical neuroscience [29]. In this article, a different more statistical approach to identifying synergy is taken, which is ultimately shown to be intimately related to information theory while offering significant advantages beyond classical entropy-based quantities.

### 1.2. Model-Free Interactions and the Inverse Ising Problem

In 1957, E.T. Jaynes famously showed that statistical equilibrium mechanics can be seen as a maximum entropy solution to the inverse problem of constructing a probability distribution that best reproduces a sample distribution [30]. More precisely, the equilibrium dynamics of the (inhomogeneous, or glass-like) generalised Ising model with interactions up to the *n*th order arise naturally as the maximum entropy distribution compatible with a dataset after observing the first *n* moments among binary variables. This means that in order to reproduce the moments in the data in a maximally non-committal way, it is necessary to introduce higher-order interactions, i.e., terms that involve more than two variables, in the description of the system. Fitting such a generalised Ising model to data is nontrivial; while the log-likelihood of the Ising model is concave in the the coupling parameters, the cost of evaluating it is exponential in the total number of variables *N*, which is often intractable in practice [31]. In [32], the authors introduced an estimator of model-free interactions (MFIs) that exactly coincides with the solution to the inverse generalised Ising problem. Moreover, the cost of estimating all *n*th-order model-free interactions among *N* variables from *M* observations scales as OM·Nn=O(MNn) (i.e., polynomially) in the total system size *N*. However, this is true only when sufficient data is available. With limited data, certain interactions might require inferring the conditional dependencies from the data, which in the worst case scales exponentially in *N* again. The definition of MFIs offered in [32] seems to be a general one; in addition to offering a solution to the inverse generalised Ising problem, MFIs are expressible in terms of average treatment effects (ATEs) or regression coefficients. Throughout this article, the general term ‘MFI’ is used, and may be read simply as referring to the maximum entropy or Ising interaction.

### 1.3. Outline

In Section 2.1, the definition of the MFIs is stated along with a number of their properties. To explicitly link the MFIs to information theory, a redefinition of mutual information in terms of Möbius inversions is provided in Section 2.2, which is then linked to a similar redefinition of the MFIs in Section 3.1 and Section 3.2. A definition in terms of Möbius inversions naturally leads to dual definitions of all objects, which are subsequently explored in Section 3.3. Then, in Section 4, simple fundamental examples are used to demonstrate that MFIs can differentiate distributions that entropy-based quantities cannot. Finally, the results are summarised and reflected upon in Section 5.

## 2. Background

### 2.1. Model-Free Interactions

We start by re-defining the interactions introduced in [32]. We define the isolated effect (or 1-point interaction) $I_{i}^{\left(Y\right)}$ of a variable $X_{i}\in X$ on an observable *Y* as
(1)$$I_{i}^{\left(Y\right)}=\frac{\partial Y}{\partial X_{i}}{|}_{\underline{X}=0}\;\;\;,\;\;\;\underline{X}=X\backslash \left\{X_{i}\right\}$$
where the effect of $X_{i}$ on *Y* is isolated by conditioning on all other variables being zero. This expression is well-defined, as the restriction of a derivative is the derivative of the restriction. A pair of variables $X_{i}$ and $X_{j}$ has a 2-point interaction $I_{ij}^{\left(Y\right)}$ when the value of $X_{j}$ changes the 1-point interaction of $X_{i}$ on *Y*:(2)$$I_{ij}^{\left(Y\right)}=\frac{\partial I_{i}^{\left(Y\right)}}{\partial X_{j}}{|}_{\underline{X}=0}=\frac{\partial^{2}Y}{\partial X_{j}\partial X_{i}}{|}_{\underline{X}=0}\;\;\;,\;\;\;\underline{X}=X\backslash \left\{X_{i},X_{j}\right\}$$ A third variable $X_{k}$ can modulate this 2-point interaction through what we call a 3-point interaction, $I_{ijk}^{\left(Y\right)}$:(3)$$I_{ijk}^{\left(Y\right)}=\frac{\partial I_{ij}^{\left(Y\right)}}{\partial X_{k}}{|}_{\underline{X}=0}=\frac{\partial^{3}Y}{\partial X_{k}\partial X_{j}\partial X_{i}}{|}_{\underline{X}=0}\;\;\;,\;\;\;\underline{X}=X\backslash \left\{X_{i},X_{j},X_{k}\right\}$$This process of taking derivatives with respect to an increasing number of variables can be repeated to define *n*-point interactions.

**Definition** **1** (*n*-point interaction with respect to outcome *Y*)**.**
*Let p be a probability distribution over a set X of random variables $X_{i}$ and let Y be a function Y:X→R. Then, the n-point interaction $I_{X_{1}\cdots X_{n}}$ between variables $\{X_{1},\ldots ,X_{n}\}\subseteq X$ is provided by*

(4)
$$\begin{array}{c} I_{X_{1}\ldots X_{n}}^{\left(Y\right)}=\frac{\partial^{n}Y\left(X\right)}{\partial X_{1}\ldots \partial X_{n}}{|}_{\underline{X}=0} \end{array}$$

*where $\underline{X}=X\backslash \left\{X_{1},\ldots X_{n}\right\}$.*


This definition of interaction makes explicit the fact that interactions are defined with respect to some outcome. The authors of [32] refer to the interactions from Definition 1 as *additive*, which they distinguish from *multiplicative* interactions. However, when the outcome is chosen to be the log of the joint distribution p(X) over all variables *X*, then the additive and multiplicative interactions are equivalent and simply related through a logarithm [32]. Setting the outcome to be logp(X) has other nice properties as well. First, while probabilities are restricted to the non-negative reals, a log-transformation removes this restriction and makes the outcome and subsequent interactions take both positive and negative values, which can have different interpretations. Second, it is this outcome that makes the interactions interpretable as maximum entropy interactions, as they exactly coincide with Ising interactions. Finally, this can be considered the most general outcome possible, as all marginal and conditional probabilities are encoded in this joint distribution. This leads to the following definition of a model-free interaction.

**Definition** **2** (model-free *n*-point interaction between binary variables)**.**
*A model-free n-point interaction (MFI) is an n-point interaction between binary random variables with respect to the logarithm of their joint probability*

(5)
$$\begin{array}{c} I_{X_{1}\ldots X_{n}}I_{X_{1}\ldots X_{n}}^{\left(\log p(X)\right)}=\frac{\partial^{n}\log p\left(X\right)}{\partial X_{1}\ldots \partial X_{n}}{|}_{\underline{X}=0} \end{array}$$

*where X_=X\{X1,…Xn}.*


If the variables $X_{i}\in X$ are binary, then a definition for a derivative with respect to a binary variable is needed.

**Definition** **3** (derivative of a function with respect to a binary variable)**.**
*Let $f:B^{n}\to R$ be a real-valued function of a set X of n binary variables, labelled as $X_{i}$, 1≤i≤n. Then, the derivative operator with respect to $X_{i}$ acts on f(X) as follows:*

(6)
$$\begin{array}{c} \frac{\partial }{\partial X_{i}}f\left(X\right)=f\left(X_{i}=1,X\backslash X_{i}\right)-f\left(X_{i}=0,X\backslash X_{i}\right) \end{array}$$

*The linearity of the derivative operator then immediately and uniquely defines the higher-order derivatives.*


Using this definition, the n-point interactions become model-free in the sense that they are ratios of probabilities that do not involve the functional form of the joint probability distribution. For example, writing $X_{ijk}=\left(a,b,c\right)$ for $\left(X_{i}=a,X_{j}=b,X_{k}=c\right)$, the first three orders can be written out as follows (recall that the notation $\frac{\partial }{\partial X_{i}}$ here refers to the derivative operator from Definition 3): (7)Ii=∂logp(X)∂Xi|X_=0=logpXi=1∣X_=0pXi=0∣X_=0(8)Iij=∂2logp(X)∂Xj∂Xi|X_=0=logpXij=(1,1)∣X_=0pXij=(0,1)∣X_=0pXij=(0,0)∣X_=0pXij=(1,0)∣X_=0 Iijk=∂3logp(X)∂Xk∂Xj∂Xi|X_=0=(9)logpXijk=(1,1,1)∣X_=0pXijk=(0,0,0)∣X_=0pXijk=(1,0,0)∣X_=0pXijk=(0,1,1)∣X_=0 ×pXijk=(0,1,0)∣X_=0pXijk=(1,0,1)∣X_=0pXijk=(0,0,1)∣X_=0pXijk=(1,1,0)∣X_=0
where Bayes’ rule is used to replace joint probabilities with conditional probabilities. This definition of interaction has the following properties:It is symmetric in terms of the variables, as $I_{S}=I_{\pi (S)}$ for any set of variables *S* and any permutation π.Conditionally independent variables do not interact: $X_{i}\;⊥\;\;\;⊥X_{j}|\underline{X}\Rightarrow I_{ij}=0$.If $\underline{X}=\emptyset$, the definition coincides with that of a log-odds ratio, which has already been considered as a measure of interaction in, e.g., [33,34].The interactions are model-free; no knowledge of the functional form of p(X) is required, and the probabilities can be directly estimated from i.i.d. samples.The MFIs are exactly the Ising interactions in the maximum entropy model after observing moments of the data. This can be readily verified by setting
$$p\left(s\right)=Z^{-1}\exp\left(\sum_{n}\sum_{i_{1},\ldots ,i_{n}}J_{i_{1}\ldots i_{n}}s_{i_{1}}\ldots s_{i_{n}}\right)$$
and using Definition 2.

Furthermore, in Section A.2 the following two useful properties are introduced and proved:An *n*-point interaction can only be non-zero if all *n* variables are in each other’s minimal Markov blanket.If $\underline{X}$ does not include the full complement of the interacting variables, the bias this induces in the estimate of the interaction is proportional to the pointwise mutual information of states where the omitted variables are 0.

### 2.2. Mutual Information as a Möbius Inversion

The definition of an *n*-point interaction as a derivative of a derivative is reminiscent of Gregory Bateson’s view of information as a *difference which makes a difference* [35]; however, the relationship between information theory and model-free interactions rests on more than a linguistic coincidence. It turns out that interactions and information are generalised derivatives of similar functions on Boolean algebras. To see this, consider the definition of pairwise mutual information and its third-order generalisation: (10)MI(X,Y)=H(X)−H(X∣Y)(11)=H(X)+H(Y)−H(X,Y)(12)MI(X,Y,Z)=MI(X,Y)−MI(X,Y∣Z) =H(X)+H(Y)+H(Z) −H(X,Y)−H(X,Z)−H(Y,Z)+H(X,Y,Z)

Note that all MI-based quantities can be written thusly as sums of marginal entropies of subsets of the set of variables. Given a finite set of variables *S*, its powerset P(S) can be assigned a partial ordering as follows:(13)$$\begin{array}{c} a\leq b\Leftrightarrow a\subseteq b\;\;\;\;\forall \;a,b\in P(S) \end{array}$$

This poset P=(P(S),⊆) is called a Boolean algebra, and because each pair of sets has a unique supremum (their union) and infimum (their intersection), it is a lattice. This lattice structure is visualised for two and three variables in Figure 1. In general, the lattice of an *n*-variable Boolean algebra forms an *n*-cube. Furthermore, for any finite *n*, the *n*-variable Boolean algebra forms a bounded lattice, which means that it has a *greatest* element, denoted as $\hat{1}$, and a *least* element, denoted as $\hat{0}$.

On a poset *P*, we define the Möbius function $\mu_{P}:P\times P\to R$ as
(14)$$\begin{array}{c} \mu_{P}\left(x,y\right)=\left\{\begin{array}{cc} 1 & \mathrm{if}\;x=y\\ -\sum_{z:x\leq z<y}\mu_{P}\left(x,z\right) & \mathrm{if}\;x<y\\ 0 & \mathrm{otherwise} \end{array}\right. \end{array}$$

This function type makes $\mu_{P}$ an element of the *incidence algebra* of *P*. In fact, μ is the inverse of the zeta function ζ:ζ(x,y)=1iffx≤y, and 0 otherwise. On a Boolean algebra, such as a powerset ordered by inclusion, the Möbius function takes the simple form $\mu \left(x,y\right)={\left(-1\right)}^{|x|-|y|}$ [36,37]. This definition allows the mutual information among a set of variables τ to be written as follows [38,39]:(15)MI(τ)=(−1)∣τ∣−1∑η≤τμP(η,τ)H(η)(16)=∑η≤τ(−1)∣η∣+1H(η)
where *P* is the Boolean algebra with $\tau =\hat{1}$ and H(η) is the marginal entropy of the set of variables η. Indeed, this coincides with Equation (11) for τ={X,Y} and with Equation (13) for τ={X,Y,Z}. Equation (15) is a convolution known as a Möbius inversion.

**Definition** **4**(Möbius inversion over a poset, Rota (1964) [37])**.**
*Let P be a poset (S,≤), let μ:P×P→R be the Möbius function from Equation* (Equation 14)*, and let g:P→R be a function on P. Then, the function*
(17)$$\begin{array}{c} f\left(y\right)=\sum_{x\leq y}\mu_{P}\left(x,y\right)g\left(x\right) \end{array}$$
*is called the Möbius inversion of g on P. Furthermore, this equation can be inverted to yield*
(18)$$\begin{array}{c} f\left(y\right)=\sum_{x\leq y}\mu_{P}\left(x,y\right)g\left(x\right)\Leftrightarrow g\left(y\right)=\sum_{x\leq y}f\left(x\right) \end{array}$$

The Möbius inversion is a generalisation of the derivative to posets. If P=(N,≤), Equation (Equation 18) is just a discrete version of the fundamental theorem of calculus [36]. Equation (Equation 18) additionally implies that we can express joint entropy as a sum over mutual information:(19)$$\begin{array}{c} H\left(\tau \right)={\left(-1\right)}^{|\tau |-1}\sum_{\eta \leq \tau }MI\left(\eta \right) \end{array}$$For example, in the case of three variables,
(20)$$\begin{array}{c} H(X,Y,Z)=MI(X,Y,Z)+MI(X,Y)+MI(X,Z)+MI(Y,Z)+H(X)+H(Y)+H(Z) \end{array}$$

Instead of starting with entropy, we could start with a quantity known as surprisal, or self-information, defined as the negative log probability of a certain state or realisation:(21)$$\begin{array}{cc} S(X=x)= & -\log p(X=x) \end{array}$$
Surprisal plays an important role in information theory; indeed, the expected surprisal across all possible realisations X=x is the entropy of the variable *X*:(22)$$\begin{array}{c} E_{X}\left[S\left(X=x\right)\right]=H\left(X\right) \end{array}$$
As we are often interested in the marginal surprisal of a realisation X=x summed over *Y*, we can write this explicitly as
(23)$$\begin{array}{c} \log p\left(x;Y\right):=\sum_{y}\log p\left(x,y\right) \end{array}$$
With this, consider the Möbius inversion of the marginal surprisal over the lattice *P*: (24)$$\begin{array}{c} \mathrm{pmi}\left(T=\tau \right):={\left(-1\right)}^{|\tau |}\sum_{\eta \leq \tau }\mu_{P}\left(\eta ,\tau \right)\log p\left(\eta ;\tau \backslash \eta \right) \end{array}$$
This is a generalised version of the pointwise mutual information, which is usually defined on just two variables:(25)pmi(X=x,Y=y)=log(x,y;∅)−log(x;Y)−log(y;X)+log(∅;X,Y)(26)=logp(x,y)p(x)p(y)
**Summary**
*Mutual information is the Möbius inversion of marginal entropy*.*Pointwise mutual information is the Möbius inversion of marginal surprisal*.

## 3. Interactions and Their Duals

### 3.1. MFIs as Möbius Inversions

With mutual information defined in terms of Möbius inversions, the same can be done for the model-free interactions. Again, we start with (negative) surprisal. However, on Boolean variables a state is just a partition of the variables into two sets: one in which the variables are set to 1, and another in which they are set to 0. That means that the surprisal of observing a particular state is completely specified by which variables X⊆Z are set to 1 while keeping all other variables Z\X at 0, which can be written as
(27)$$\begin{array}{cc} S_{X;Z} & :=\log p(X=1,Z\backslash X=0) \end{array}$$

**Definition** **5** (interactions as Möbius inversions)**.**
*Let p be a probability distribution over a set T of random variables and let P=(P(τ),⊆), the powerset of a set τ⊆T ordered by inclusion. Then, the interaction I(τ;T) among variables τ is provided by*

(28)I(τ;T):=∑η≤τμP(η,τ)Sη;T(29)=∑η≤τ(−1)∣η∣−∣τ∣logp(η=1,T\η=0)



For example, when τ contains a single variable X⊆T, then
(30)I({X};T)=μP({X},{X})S{X};T+μP(∅,{X})S∅;T(31)=logp(X=1,T\X=0)p(X=0,T\X=0)
which coincides with the 1-point interaction in Equation (7). Similarly, when τ contains two variables τ={X,Y}⊆T, then
(32)I({X,Y};T)=μP({X,Y},{X,Y})S{X,Y};T+μP({X},{X,Y})S{X};T +μP({Y},{X,Y})S{Y};T+μP(∅,{X,Y})S∅;T(33)=logp(X=1,Y=1,T\{X,Y}=0)p(X=0,Y=0,T\{X,Y}=0)p(X=1,Y=0,T\{X,Y}=0)p(X=0,Y=1,T\{X,Y}=0)
which coincides with the 2-point interaction in Equation (8). In fact, this pattern holds in general.

**Theorem** **1** (equivalence of interactions)**.**
*The interaction I(τ,T) from Definition 5 is the same as the model-free interaction $I_{\tau }$ from Definition 2, that is, for any set of variables τ⊆T it is the case that*

(34)
$$\begin{array}{c} I\left(\tau ,T\right)=I_{\tau } \end{array}$$



**Proof.** We have to show that
(35)∑η≤τ(−1)∣η∣−∣τ∣logp(η=1,T\η=0)=∂nlogp(T)∂τ1…∂τn|T_=0Both sides of this equation are sums of ±logp(s), where *s* is some binary string; thus, we have to show that the same strings appear with the same sign.First, note that the Boolean algebra of sets ordered by inclusion (as in Figure 1) is equivalent to the poset of binary strings where for any two strings *a* and *b*, a≤b⇔a∧b=a. The equivalence follows immediately upon setting each element a∈P(S) to the string where a=1 and S\a=0. This map is one-to-one and monotonic with respect to the partial order, as A⊆B⇔A∩B=A. This means that Definition 5 can be rewritten as a Möbius inversion on the lattice of Boolean strings S=(B|τ|,≤) (shown for the three-variable case on the left side of Figure 2):
(36)I(τ;T)=∑s≤1^SμS(s,1^S)logp(τ=s,T\τ=0)
Note that for any pair (α,τ) where α⊆τ with respective string representations (s,t)∈B|τ|×B|τ|, we have the following:
(37)|τ|−|α|=∑i(t∧¬s)i
Thus, we can write
(38)$$I\left(\tau ;T\right)=\sum_{s\leq {\hat{1}}_{S}}{\left(-1\right)}^{\sum \neg s}\log p\left(\tau =s,T\backslash \tau =0\right)$$
To see that this exactly coincides with Definition 2, we can define a map
(39)$$e_{i,s}^{\left(n\right)}:F_{B^{n}}\to F_{B^{n-1}}$$
where $F_{B^{n}}$ is the set of functions from *n* Boolean variables to R. This map is defined as
(40)$$e_{i,s}^{\left(n\right)}:f\left(X_{1},\ldots X_{i},\ldots X_{n}\right)\mapsto f\left(X_{1},\ldots X_{i}=s,\ldots X_{n}\right)$$
With this map, the Boolean derivative of a function $f(X_{1},\ldots ,X_{n})$ (see Definition 3) can be written as
(41)$$\frac{\partial }{\partial X_{i}}f\left(X\right)=\left(e_{i,1}^{\left(n\right)}-e_{i,0}^{\left(n\right)}\right)f\left(X\right)$$
(42)$$=f\left(X_{1},\ldots ,X_{i}=1,\ldots ,X_{n}\right)-f\left(X_{1},\ldots ,X_{i}=0,\ldots ,X_{n}\right)$$
In this way, the derivative with respect to a set *S* of *m* variables becomes function composition:
(43)$$\left(\prod_{i=0}^{m}\frac{\partial }{\partial X_{S_{i}}}\right)f\left(X\right)=\left({◯}_{i=0}^{m}\left(e_{S_{i},1}^{\left(n-i\right)}-e_{S_{i},0}^{\left(n-i\right)}\right)\right)f\left(X\right)$$From this, it is clear that a term f(s) appears with a minus sign iff $e_{i,0}^{\left(n\right)}$ has been applied an odd number of times. Therefore, terms for which *s* contains an odd number of 0s receive a minus sign. This can be summarised as
(44)$$\left(\prod_{i=0}^{m}\frac{\partial }{\partial X_{S_{i}}}\right)f\left(X\right)=\sum_{s\in B^{n}}{\left(-1\right)}^{\sum \neg s}f\left(X_{S}=s,X\backslash X_{S}\right)$$
Therefore, we can write
(45)$$I_{\tau }=\sum_{s\in B^{n}}{\left(-1\right)}^{\sum \neg s}\log\left(\tau =s,T\backslash \tau =0\right)$$
The sums $\sum_{s\leq {\hat{1}}_{S}}$ and $\sum_{s\in B^{n}}$ contain exactly the same terms, meaning that Equations (Equation 38) and (Equation 45) are equal. This completes the proof.    □

Note that the structure of the lattice reveals structure in the interactions, as previously noted in [32]. On the right-hand side of Figure 2, two faces of the three-variable lattice are shaded. The green region corresponds to the 2-point interaction between the first two variables. The red region contains a similar interaction between the first two variables, except this time in the context of the third variable fixed to 1 instead of 0. This illustrates the interpretation of a 3-point interaction as the difference in two 2-point interactions ($I_{XYZ}=I_{XY|Z=1}-I_{XY|Z=0}$; note that $I_{XY|Z=0}$ is usually written as just $I_{XY}$). The symmetry of the cube reveals the three different (though equivalent) choices as to which variable to set to 1. Treating the Boolean algebra as a die, where the sides facing up are ⚀, ⚁, and ⚂, we have
(46)$$I_{XYZ}=⚀-⚅=⚁-⚄=⚂-⚃$$

As before, we can invert Definition 5 and express the surprise of observing a state with all ones in terms of interactions, as follows:(47)$$\log p\left(\tau =1,T\backslash \tau =0\right)=\sum_{\eta \leq \tau }I\left(\eta ,T\right)$$
For example, in the case where T={X,Y,Z} and τ={X,Y}
(48)$$S\left(1,1,0\right)=-\log p\left(1,1,0\right)=-I_{XY}-I_{X}-I_{Y}-I_{\emptyset }$$
which illustrates that when *X* and *Y* tend to be off ($I_{X}<0$ and $I_{Y}<0$) and *X* and *Y* tend to be different ($I_{XY}<0$), observing the state (1, 1, 0) is very surprising.

### 3.2. Categorical Interactions

Taking seriously the definition of interactions as the Möbius inversion of surprisal, one might ask what happens when surprisal is inverted over a different lattice instead of using a Boolean algebra. One example is shown in Figure 3; it corresponds to variables that can take three values—0, 1, or 2—where states are ordered by $a\leq b\Leftrightarrow \forall i:a_{i}\leq b_{i}$. To calculate interactions on this lattice, we need to know the value of Möbius functions of type μ(s,22). It can be readily verified that most Möbius functions of this type are zero, with the exceptions of μ(22,22)=μ(11,22)=1 and μ(21,22)=μ(12,22)=−1, which provide the exact terms in the interactions between two categorical variables changing from 1→2 (as defined in [32]). Calculating interactions on different sublattices with $\hat{1}=\left(21\right),\left(12\right)$ or (11) provides us with the other categorical interactions. The transitivity property of the interactions, i.e., I(X:0→2,Y:0→1)=I(X:0→1,Y:0→1)+I(X:1→2,Y:0→1), follows immediately from the structure of the lattice in Figure 3 and the alternating signs of the Möbius functions on a Boolean algebra.

### 3.3. Information and Interactions on Dual Lattices

Lattices have the property that a set with the reverse order remains a lattice; that is, if L=(S,≤) is a lattice, then $L^{\mathrm{op}}=\left(S,⪯\right)$ (where ∀a,b∈S:a⪯b⇔a≥b) is a lattice. This raises the question of what corresponds to mutual information and interaction on such dual lattices. Recognising that a poset $L=(S,\leq_{L})$ is a category C with objects *S* and a morphism f:A→B iff $B\leq_{L}A$, these become definitions in the *opposite* category $C^{\mathrm{op}}$, meaning that they define dual objects.

Let us start with mutual information. We can calculate the dual mutual information, denoted $MI^{*}$, by first noting that the dual to a Boolean algebra is another Boolean algebra, meaning that we have $\mu \left(x,y\right)={\left(-1\right)}^{\left|x|-|y\right|}$. Simply replacing *P* with $P^{\mathrm{op}}$ in Equation (15) yields
(49)$$\begin{array}{cc} MI^{*}\left(\tau \right) & =\sum_{\eta ⪯\tau }{\left(-1\right)}^{|\eta |+1}H\left(\eta \right) \end{array}$$
The dual mutual information of $\tau ={\hat{1}}_{P^{\mathrm{op}}}$ is simply $MI^{*}\left(\emptyset \right)=MI\left({\hat{1}}_{P}\right)$, that is, the mutual information among all variables. However, the dual mutual information of a singleton set *X* is
(50)MI*(X)=MI(1^P)−MI(1^P\X)(51)=Δ(X;1^P\X)
where Δ is known as the differential mutual information and describes the change in mutual information when leaving out *X* [40], i.e., when marginalising over the variable *X*. Note that a similar construction was already anticipated in [41] and that the differential mutual information has previously been used to describe information structures in genetics [39]. On the Boolean algebra of three variables {X,Y,Z}, the dual mutual information of *X* can be written out as follows: MI*(X)=μ({X},{X})H(X)+μ({X,Y},{X})H(X,Y)+(52)μ({X,Z},{X})H(X,Z)+μ({X,Y,Z},{X})H(X,Y,Z)(53)=H(X)−H(X,Y)−H(X,Z)+H(X,Y,Z)
Because Δ is the dual of mutual information, it should arguably be called the mutual co-information; however, the term co-information is unfortunately already in use to refer to normal higher-order mutual information.

To find the dual to the interactions, we start from Equation (Equation 36) and construct $S^{\mathrm{op}}=\left(B^{\left|\tau \right|},⪯\right)$, the dual to the lattice of binary strings $S=(B^{\left|\tau \right|},\leq )$. A dual interaction of variables τ⊆T is denoted as $I^{*}\left(\tau ;T\right)$, and is defined as follows: (54)$$I^{*}(\tau ;T):=\sum_{s⪯{\hat{1}}_{S^{\mathrm{op}}}}\mu_{S^{\mathrm{op}}}\left(s,{\hat{1}}_{S^{\mathrm{op}}}\right)\log p\left(\tau =s,T\backslash \tau =0\right)$$
Again, when $\tau ={\hat{1}}_{S\mathrm{op}}={\hat{0}}_{S}=\emptyset$, this is simply (${-1)}^{|\tau |}I\left({\hat{1}}_{S}\right)$, while the dual interaction of a singleton set *X* is
(55)$$I^{*}\left(X;T\right)={\left(-1\right)}^{|{\hat{1}}_{S}|-1}\left(I\left({\hat{1}}_{S};T\right)+I\left({\hat{1}}_{S}\backslash X;T\right)\right)$$
For example, on the three variable lattice in Figure 2, the dual interaction of *X* is
(56)$$I^{*}\left(X;T\right)=I\left(X,Y,Z;T\right)+I\left(Y,Z;T\right)$$
Writing $p_{ijk}$ for p(X=i,Y=j,Z=k∣T\{X,Y,Z}=0), it can be seen that this is equal to
(57)$$I^{*}\left(X;T\right)=\log\frac{p_{111}p_{100}}{p_{101}p_{110}}$$
which is similar to the 2-point interaction $I_{YZ}$ defined in Equation (8), now conditioned on X=1 instead of 0. Note the difference between dual mutual information and dual interactions here; the dual mutual information of *X* describes the effect on the mutual information from *marginalising* over *X*, whereas the dual interaction of *X* describes the effect on an interaction when *fixing*X=1. This reflects a fundamental difference between mutual information and the interactions, in that the former is an *averaged* quantity and the latter a *pointwise* quantity.

Dual interactions should probably be called co-interactions; however, to avoid confusion with the term co-information, we instead refer to them simply as dual interactions. Dual interactions are interactions that are conditioned on certain variables being 1 instead of 0. This makes them no longer equal to the Ising interactions between Boolean variables; however, there are situations in which an interaction is more interesting in the context of Z=1 instead of Z=0, for example, if *Z* is always 1 in the data under consideration.


**Summary**

*Mutual information is the Möbius inversion of marginal entropy on the lattice of subsets ordered by inclusion.*

*Differential (or conditional) mutual information is the Möbius inversion of marginal entropy on the dual lattice.*

*Model-free interactions are the Möbius inversion of surprisal on the lattice of subsets ordered by inclusion.*

*Model-free dual interactions are the Möbius inversion of surprisal on the dual lattice.*

*Dual interactions of a variable X are interactions between the other variables where X is set to 1 instead of 0.*



To summarise these relationships diagrammatically, note that surprisals form a vector space as follows. Let P(T) be the powerset of a set of variables *T* and let |P(T)|=n. This forms a lattice P=(P(T),⊆) ordered by inclusion, meaning that P(T) can be assigned a topological ordering indexed by *i* as $P\left(T\right)=\cup_{i=0}^{n}t_{i}$. Let S be the set of linear combinations of surprisals of subsets of T:(58)$$\begin{array}{c} S=\left\{\sum_{i=0}^{n}a_{i}\log p\left(t_{i}\right)|a_{i}\in R\right\} \end{array}$$
This set is assigned a vector space structure over R by the usual scalar multiplication and addition. Note that the set
(59)$$\begin{array}{c} B=\left\{\log p(t)|t\in P(T)\right\} \end{array}$$
forms a basis for this vector space, because $\sum_{i}\alpha_{i}\log p\left(t_{i}\right)=0$ has no non-trivial solutions and a span(B)=S. Only when two variables *a* and *b* are independent do we have linear dependencies in B, as it is then the case that logp(a,b)=logp(a)+logp(b). To define a map from S→R, we only need to specify its action on B and extend the definition linearly. This means that we can fully define the map ${\mathrm{eval}}_{T}:S\to R$ by specifying
(60)$$\begin{array}{cc} {\mathrm{eval}}_{T}:\log p\left(R=r\right) & \mapsto \log p(R=1,T\backslash R=0) \end{array}$$
Similarly, we can define the expectation map E:S→R as
(61)$$\begin{array}{cc} E:\log p(R=r) & \mapsto \sum_{r}p\left(R=r\right)\log p\left(R=r\right) \end{array}$$
which outputs the expected surprise over all realisations R=r. Finally, note that the Möbius inversion over a poset *P* is an endomorphism of the set $F_{P}$ of functions over *P*, defined as
(62)MP:FP→FP(63)MP:f(y)↦∑x≤yμ(x,y)f(x)
Together, these three maps ensure that the following diagram commutes:



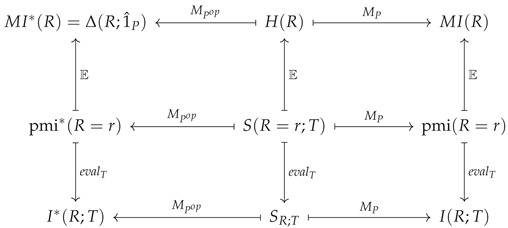



For the case where T={X,Y,Z} and R={X,Y}, this explicitly amounts to



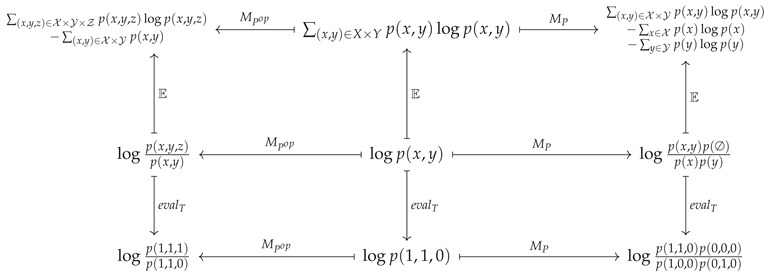



## 4. Results and Examples

While mutual information and model-free interactions are related, there are several important differences in terms of how they capture dependencies. Note, for example, that higher-order information quantities are not independent of the lower-order quantities. The mutual information of three variables is bounded by the pairwise quantities as follows:(64)$$\begin{array}{c} \;-\min\{MI(X,Y|Z),MI(Y,Z|X),MI(X,Z|Y)\}\leq MI(X,Y,Z)\leq \min\{MI(X,Y),MI(Y,Z),MI(X,Z)\} \end{array}$$
This means that there are no systems with zero pairwise mutual information and positive higher-order information. This is not true for the interactions. For example, a distribution with 3-point interactions and no pairwise interactions can trivially be constructed as $p\left(X\right)=Z^{-1}\exp\left(\sum_{ijk}J_{ijk}X_{i}X_{j}X_{k}\right)$. While this distribution has 3-point interactions with strength $J_{ijk}$ for triplets $\left\{X_{i},X_{j},X_{k}\right\}$, all pairwise interactions among $\left\{X_{i},X_{j}\right\}$ vanish when conditioning on $X_{k}=0$. In fact, any positive discrete distribution can be written as a Boltzmann distribution with an energy function that is unique up to a constant, and as such is uniquely defined by its interactions; in other words, each interaction, at any order, can be freely varied to define a unique and valid probability distribution, namely, the Boltzmann distribution of the corresponding generalised Ising model. Note that this is closely related to the fact that a class of neural networks known as restricted Boltzmann machines are universal approximators [42,43,44] and exactly (though not uniquely) encode the Boltzmann distribution of a generalised Ising model in one of their layers [31,45]. Therefore, each distribution is uniquely determined by its set of interactions, and should be distinguishable by them. This is famously not true for entropy-based information quantities, as illustrated below through several examples.

### 4.1. Interactions and Their Duals Quantify and Distinguish Synergy in Logic Gates

Under the assumption of a causal collider structure A→C←B, nonzero 3-point interactions $I_{ABC}$ can be interpreted as logic gates. A positive 3-point interaction means that the numerator in Equation (10) is larger than the denominator. Under the sufficient (though not necessary) assumption that each term in the numerator is larger than each term in the denominator, we obtain the following truth table as $I_{ABC}\to +\infty$:
* A ** B ** C *
 0   0   1  0   1   0  1   0   0  1   1   1 
which describes an XNOR gate. Let $p_{G}$ be the probability of each of the four states in the truth table for a gate G, and let $\epsilon_{G}$ be the probability of all other states. Then, the 3-point interaction of an XNOR gate can be written as
(65)$$I_{ABC}^{\mathrm{XNOR}}=\log\frac{p_{\mathrm{XNOR}}^{4}}{\epsilon_{\mathrm{XNOR}}^{4}}$$
Similarly, the truth tables of AND and OR gates imply that
(66)IABCAND=logϵANDpAND3ϵAND3pAND(67)IABCOR=logϵOR3pORϵORpOR3
If we consider equally noisy gates such that $p_{G}=p$ and $\epsilon_{G}=\epsilon$, the gates can be directly compared. Note that when a gate has a 3-point interaction *I*, its logical negation will have a 3-point interaction −I. This determines the 3-point interactions of all six non-trivial logic gates on two inputs, as summarised in Table 1. The two gates with the strongest absolute interactions, XNOR and XOR, are the only two gates that are purely synergistic, i.e., knowing only one of the two inputs provides no information about the output. This relationship to synergy holds for three-input gates as well. The three-input gate with the strongest 4-point interaction has the following truth table:
* A ** B ** C ** D *
 0   0   0   0  0   0   1   1  0   1   0   1  1   0   0   1  0   1   1   0  1   0   1   0  1   1   0   0  1   1   1   1 
This is a three-input XOR gate, i.e., D=(A+B+C)mod2, and is again maximally synergistic, as observing only two of the three inputs provides zero bits of information on the output. Setting this maximum 4-point interaction to *I*, the three-input OR and AND gates receive a 4-point interaction I/4; thus, the hierarchies of interaction and synergy continue to match.

The 3-point interactions are able to separate most two-input logic gates by sign or value, leaving only AND∼NOR and OR∼NAND. Mutual information has less resolving power. Assuming a uniform distribution over all four allowed states from a gate’s truth table, a brief calculation yields
(68)MIOR(A,B,C)=MIAND(A,B,C)=MINOR(A,B,C)=MINAND(A,B,C) =−log33/44−1≈−0.189(69)MIXOR(A,B,C)=MIXNOR(A,B,C)=−1
That is, higher-order mutual information resolves strictly fewer logical gates by value and none by sign. In fact, the higher-order mutual information of a logic gate can never be positive, because it is bounded from above by the minimum of the pairwise mutual information, which is always zero for the pair of inputs. Because all entropy-based quantities inherit the degeneracy summarised in Table 2, neither the mutual information nor its dual can increase the resolving power (see Table 3).

The logic gate interactions and their duals are summarised in Table 3, where it can be seen that neither $I_{C}^{*G}=I_{ABC}^{G}+I_{AB}^{G}$ nor $I_{A}^{*G}$ improve the resolution beyond that of the 3-point interaction. However, the 3-point interaction requires $2^{3}=8$ probabilities to achieve this resolving power, whereas $I_{C}^{*G}=\frac{p_{111}p_{001}}{p_{101}p_{011}}$ achieves the same resolving power with just four probabilities.

However, note that because of a difference in sign convention dual mutual information is a difference between two mutual information quantities, while dual interactions are a sum of two interactions. Based on this, we can consider the difference of two interactions and define a new quantity $J_{A}^{*G}=I_{ABC}^{G}-I_{BC}^{G}$. We refer to this as a *J*-interaction. When the MFIs are interpreted in the context of an energy-based model, such as an Ising model or a restricted Boltzmann machine, then the interactions have dimensions of energy, meaning that the *J*-interactions correspond to the difference in the energy contribution between a triplet and a pair. These *J*-interactions of the input nodes *A* and *B* assign a different value to each logic gate G, and the symmetric *J*-interaction ${\bar{J}}^{*G}=J_{A}^{*G}J_{B}^{*G}J_{C}^{*G}$, analogous to the symmetric deltas from [40], inherits the perfect resolution from $J_{A}^{*G}$.

Note that while $J_{A}^{*G}=J_{B}^{*G}$ both have perfect resolution, $J_{C}^{*G}=I_{ABC}^{G}-I_{AB}^{G}$ does not improve the resolution beyond that of the 3-point interaction. This results from the fact that in logic gates we have $I_{ABC}^{G}=-2I_{AB}^{G}$, meaning that $I_{ABC}^{G}$ and $I_{AB}^{G}$ contain the same information. To see this, note that
(70)$$I_{ABC}^{G}+2I_{AB}^{G}=\log\frac{p_{111}p_{001}p_{110}p_{000}}{p_{101}p_{011}p_{010}p_{100}}$$
Because the logic gates are symmetric in their inputs, i.e., $\forall i,j\;p_{ijk}=p_{jik}$, this can be rewritten as
(71)$$I_{ABC}^{G}+2I_{AB}^{G}=\log\frac{\left(p_{111}p_{110}\right)\left(p_{001}p_{000}\right)}{\left(p_{101}p_{100}\right)\left(p_{101}p_{100}\right)}$$
Each of these terms in brackets has the form $\left(p_{ij1}p_{ij0}\right)$. Because these are two contradicting states, this product reduces to ϵp regardless of the truth table of G:(72)$$I_{ABC}^{G}+2I_{AB}^{G}=\log\frac{\epsilon^{2}p^{2}}{\epsilon^{2}p^{2}}=0$$
Note that this pattern could already be observed in Table 3, though it was not yet explained.

Thus, the *J*-interactions of the input nodes uniquely assign a value to each gate proportional to the synergy of its logic. The hierarchy is $J_{A}^{*\mathrm{XNOR}}>J_{A}^{*\mathrm{NOR}}>J_{A}^{*\mathrm{AND}}$, which is mirrored for the respective logical complements. XNOR is indeed the most synergistic, while NOR is more synergistic than AND with respect to observing a 0 in one of the inputs; in a NOR gate, a 0 in the input provides no information on the output, while it completely fixes the output of an AND gate. Because the interactions are defined in a context of 0s, they order the synergy accordingly.

### 4.2. Interactions Distinguish Dynamics and Causal Structures

To illustrate how different association metrics reflect the underlying causal dynamics, consider data generated from a selection of three-node causal DAGs as follows. On a given DAG G, first denote the set of nodes without parents, the orphan nodes, by $S_{0}$. Each orphan node in $S_{0}$ receives a random value drawn from a Bernoulli distribution, i.e., P(X=1)=p and P(X=0)=1−p. Next, denote the set of children of orphan nodes as $S_{1}$. Each node in $S_{1}$ is then set to either the product of its parent nodes (for *multiplicative* dynamics) or the mean of its parent nodes (for *additive* dynamics), plus some zero-mean Gaussian noise with variance $\sigma^{2}$. Note that for the fork and the chain this simply amounts to a noisy copying operation. All nodes are then rounded to a 0 or 1. A set $S_{2}$ is then defined as the set of all children of nodes in $S_{1}$, and these receive values using the same dynamics as before. As long as the causal structure is acyclic, this algorithm terminates on a set of nodes $S_{i}$ that has no children. For example, the chain graph A→B→C has $S_{0}=\left\{A\right\}$, $S_{1}=\left\{B\right\}$, $S_{2}=\left\{C\right\}$, and $S_{3}=\emptyset$, at which point the updating terminates.

Figure 4 shows the results for four different DAGs with multiplicative and additive dynamics (though these are the same for forks and chains). The six different dynamics are represented in four different DAGs, two different (Pearson) correlations, four different partial correlations, and two different mutual information structures, which means that each of these descriptions is degenerate in some of the dynamics. The shown pairwise partial correlations are the correlations among the residuals after a linear regression against the third variable. Because this is similar to conditioning on the third variable, it is somewhat analogous to the MFIs; in fact, when the variables are multivariate normal the partial correlations are encoded in the inverse covariance matrix and are equivalent to pairwise Ising interactions [31]. Indeed, it can be seen that the partial correlations are somewhat able to disentangle direct effects from indirect effects, although they fail to distinguish additive from multiplicative dynamics. Note that only the sign of the association and its significance are represented, as the precise value depends on the noise level $\sigma^{2}$. The rightmost column shows that the MFIs assign a unique association structure to each of the dynamics, distinguish between direct and indirect effects, and reveal multiplicative dynamics as a 3-point interaction while identifying additive dynamics as a purely pairwise process. Finally, note that both the partial correlation and the MFIs assign a negative association to the parent nodes in a collider structure. This reflects that two nodes become dependent when conditioned on a common effect (cf. Berkson’s paradox), a phenomenon already found in partial correlations of metabolomic data in [46]. The mutual information is affected by Berkson’s paradox as well, revealed through the negative three-point mutual information. This negative three-point is a direct effect from conditioning on the common effect *C*, as on colliders MI(A,B,C)=MI(A,B)−MI(A,B∣C)=−MI(A,B∣C), because the mutual information among the independent inputs *A* and *B* vanishes by definition.

### 4.3. Higher-Order Categorical Interactions Distinguish Dyadic and Triadic Distributions

That the interactions have such resolving power over distributions of binary variables is perhaps not very surprising in light of the universality of RBMs with respect to this class of distributions. More surprisingly, their resolving power extends to the case of categorical variables. In [47], the authors introduced two distributions, the dyadic and triadic distributions, which are indistinguishable by almost all commonly used information measures (i.e., Shannon, Renyi(2), residual, and Tsallis entropy, co-information, total correlation, CAEKL mutual information, interaction information, Wyner, exact, functional, and MSS common information, perplexity, disequilibrium, and LMRP and TSE complexities).

**Figure 4 entropy-25-00648-f004:**
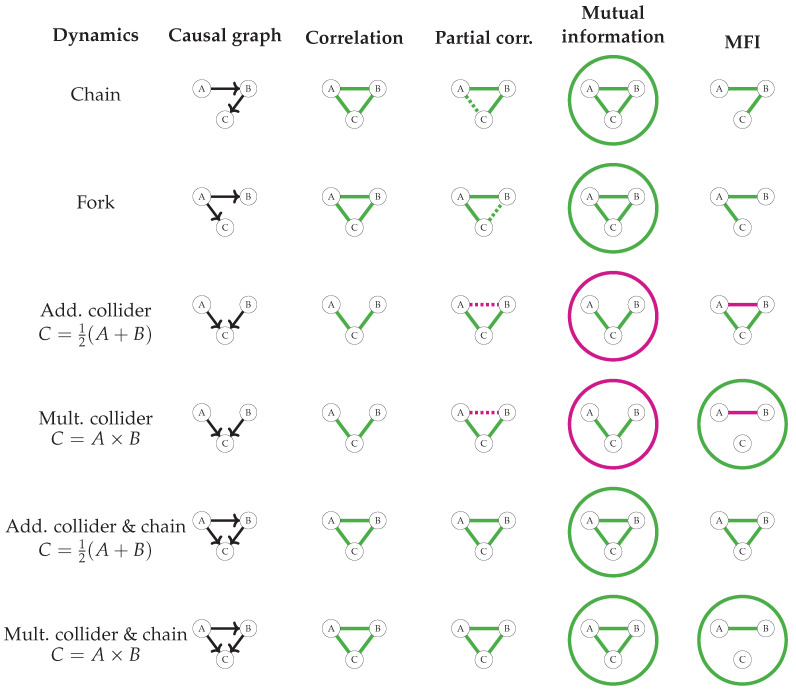
Different causal dynamics lead to different association metrics. Green edges denote positive values, red edges denote negative values, circles denote a three-point quantity, and dashed lines show edges with marginal significance (depending on $\sigma^{2}$). Correlations and mutual information cannot distinguish between most dynamics, and while partial correlation can identify the correct pairwise relationships for certain noise levels, it falls short of distinguishing additive from multiplicative dynamics. Only MFIs can distinguish between all six scenarios and reveal the combinatorial effect of the multiplicative dynamics as a 3-point interaction. See Section A.3 for the simulation parameters and raw numbers. This figure is reproduced with permission from the author of [48].

The two distributions are defined on three variables, each taking a value in a four-letter alphabet {0,1,2,3}. The joint probabilities are summarised in Table 4. To construct the distributions, each category is represented as a binary string ({0,1,2,3}→{00,01,10,11}), leading to new variables $\left\{X_{0},X_{1},Y_{0},Y_{1},Z_{0},Z_{1}\right\}$. The dyadic distribution is constructed by linking these new variables with pairwise rules $X_{0}=Y_{1},Y_{0}=Z_{1},Z_{0}=X_{1}$, while the triadic distribution is constructed with triplet rules $X_{0}+Y_{0}+Z_{0}=0\;\mathrm{mod}\;2$ and $X_{1}=Y_{1}=Z_{1}$. The resulting binary strings are then reinterpreted as categorical variables to produce Table 4.

The authors of [47] found that no Shannon-like measure can distinguish between the two distributions, and argued that the partial information decomposition, which is different for the two distributions, is not a natural information measure, as it has to single out one of the variables as an output. To calculate model-free categorical interactions between the variables, we can set the probabilities of the states in Table 4 uniformly to p=(1−(64−8)ϵ)/8 and those of the other states to ϵ (i.e., a normalised uniform distribution over legal states). There are a total of $6^{3}=216$ interactions such that $x_{1}>x_{0},\;y_{1}>y_{0},\;z_{1}>z_{0}$. Each of these can be written as
(73)$$\begin{array}{cc} & I_{XYZ}\left(x_{0}\to x_{1};y_{0}\to y_{1};z_{0}\to z_{1}\right)=\\ \log & \frac{p\left(X=x_{1},Y=y_{1},Z=z_{1}|\underline{X}=0\right)}{p\left(X=x_{0},Y=y_{0},,Z=z_{0}|\underline{X}=0\right)}\frac{p\left(X=x_{1},Y=y_{0},Z=z_{0}|\underline{X}=0\right)}{p\left(X=x_{0},Y=y_{1},,Z=z_{1}|\underline{X}=0\right)}\\ \times & \frac{p\left(X=x_{0},Y=y_{1},Z=z_{0}|\underline{X}=0\right)}{p\left(X=x_{1},Y=y_{0},,Z=z_{1}|\underline{X}=0\right)}\frac{p\left(X=x_{0},Y=y_{0},Z=z_{1}|\underline{X}=0\right)}{p\left(X=x_{1},Y=y_{1},,Z=z_{0}|\underline{X}=0\right)} \end{array}$$

Of particular interest here are the two quantities $I_{XYZ}\left(0\to 3;0\to 3;0\to 3\right)$ and ${\bar{I}}_{XYZ}=\sum_{x_{0},x_{1},y_{0},y_{1},z_{0},z_{1}}I_{XYZ}\left(x_{0}\to x_{1};y_{0}\to y_{1};z_{0}\to z_{1}\right)$, where the sum is over all values such that $x_{1}>x_{0},y_{1}>y_{0},z_{1}>z_{0}$, as all possible pairs necessarily sum to zero because $I_{XYZ}\left(x_{0}\to x_{1};y_{0}\to y_{1};z_{0}\to z_{1}\right)=-I_{XYZ}\left(x_{1}\to x_{0};y_{0}\to y_{1};z_{0}\to z_{1}\right)$. For the dyadic distribution, we have
(74)$$I_{XYZ}^{\mathrm{Dy}}\left(0\to 3;0\to 3;0\to 3\right)=\log\frac{p\epsilon^{3}}{p\epsilon^{3}}=0,$$
while for the triadic distribution we have
(75)$$I_{XYZ}^{\mathrm{Tri}}\left(0\to 3;0\to 3;0\to 3\right)=\log\frac{\epsilon^{4}}{p\epsilon^{3}}=\log\frac{\epsilon }{p}$$
Thus, this particular 3-point interaction is zero for the dyadic distribution and negative for the triadic distribution. The sum over all three points (see Section A.4 for details) is provided by
(76)I¯XYZDy=log1=0(77)I¯XYZTri=64logϵp

That is, the additively symmetrised 3-point interaction is zero for the dyadic distribution and strongly negative for the triadic distribution. These two distributions, which are indistinguishable in terms of their information structure, are distinguishable by their model-free interactions, which accurately reflect the higher-order nature of the triadic distribution.

## 5. Discussion

In this paper, we have related the model-free interactions introduced in [32] to information theory by defining them as Möbius inversions of surprisal on the same lattice that relates mutual information to entropy. We then invert the order of the lattice and compute the order-dual to the mutual information, which turns out to be a generalisation of differential mutual information. Similarly, the order-dual of interaction turns out to be interaction in a different context. Both the interactions and the dual interactions are able to distinguish all six logic gates by value and sign. Moreover, their absolute strength reflects the synergy within the logic gate. In simulations, the interactions were able to perfectly distinguish six kinds of causal dynamics that are partially indistinguishable to Pearson/partial correlations, causal graphs, and mutual information. Finally, we considered dyadic and triadic distributions constructed using pairwise and higher-order rules, respectively. While these two distributions are indistinguishable in terms of their Shannon information, they have different categorical MFIs that reflect the order of the construction rules.

One might wonder why the interactions enjoy this advantage over entropy-based quantities. The most obvious difference is that the interactions are defined in a pointwise way, i.e., in terms of the surprisal of particular states, whereas entropy is the expected surprisal across an ensemble of states. Furthermore, the MFIs can be interpreted as interactions in an Ising model and as effective couplings in a restricted Boltzmann machine. As both these models are known to be universal approximators with respect to positive discrete probability distributions, the MFIs should be able to characterise all such distributions. What is not immediately obvious is that the kinds of interactions that characterise a distribution should reflect properties of that distribution, such as the difference between direct and indirect effects and the presence of higher-order structure. However, in the various examples covered in this manuscript the interactions turn out to intuitively align with properties of the process used to generate the data. While the stringent conditioning on variables not considered in the interaction might make it tempting to interpret an MFI as a causal or interventional quantity, it is important to be very careful when doing this. Assigning a causal interpretation to statistical inferences, whether in Pearl’s graphical do-calculus [49] or in Rubin’s potential outcomes framework [50], requires further (often untestable) assumptions and analysis of the system in order to determine whether a causal effect is identifiable and which variables to control for. In contrast, an MFI is simply defined by conditioning on all observed variables, makes no reference to interventions or counterfactuals, and does not specify a direction of the effect. While in a controlled and simple setting the MFIs can be expressed in terms of causal average treatment effects [32], a causal interpretation is not justifiable in general.

Moreover, the stringency in the conditioning might worry the attentive reader. Estimating logp(X=1,Y=1,T=0) directly from data means counting states such as $\left(X,Y,T_{1},T_{2},\ldots ,T_{N}\right)=\left(1,1,0,0,\ldots 0\right)$, which for sufficiently large *N* are rare in most datasets. Section A.1 shows how to use the causal graph to construct Markov blankets, making such estimation tractable when full conditioning is too stringent. In an upcoming paper, we address this issue by estimating the graph of conditional dependencies, allowing for successful calculation of MFIs up to the fifth order in gene expression data.

One major limitation of MFIs is that they are only defined on binary or categorical variables, whereas many other association metrics are defined for ordinal and continuous variables as well. As states of continuous variables no longer form a lattice, it is hard to see how the definition of MFIs could be extended to include these cases.

Finally, it is worth noting that the structure of different lattices has guided much of this research. That Boolean algebras are important in defining higher-order structure is not surprising, as they are the stage on which the inclusion–exclusion principle can be generalised [36]. However, it is not only their order-reversed duals that lead to meaningful definitions; completely unrelated lattices do as well. For example, the Möbius inversion on the lattice of ordinal variables from Figure 3 and the redundancy lattices in the partial information decomposition [28] both lead to new and sensible definitions of information-theoretic quantities. Furthermore, the notion of Möbius inversion has been generalised to a more general class of categories [51], of which posets are a special case. A systematic investigation of information-theoretic quantities in this richer context would be most interesting.

## Figures and Tables

**Figure 1 entropy-25-00648-f001:**
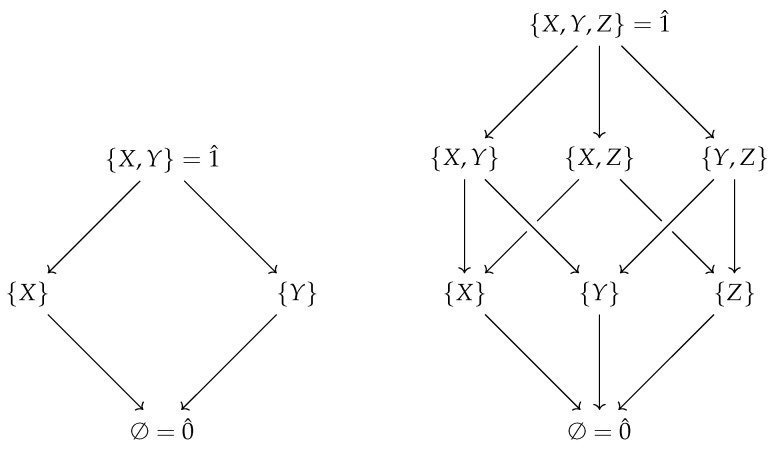
The lattices associated with P({X,Y}) (**left**) and P({X,Y,Z}) (**right**) ordered by inclusion. An arrow b→a indicates a<b.

**Figure 2 entropy-25-00648-f002:**
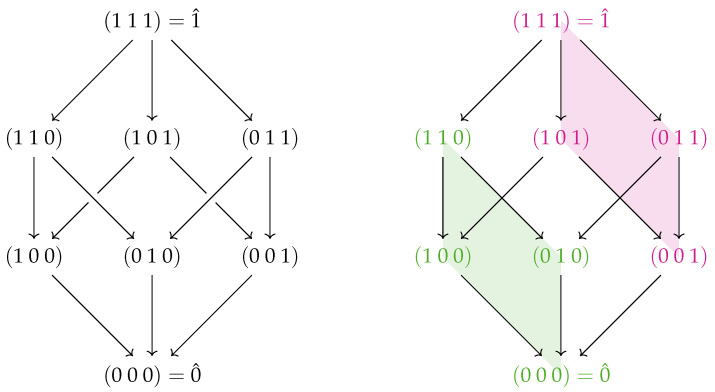
(**Left**) The lattice associated with P({X,Y,Z}) ordered by inclusion as binary strings. Equivalently, the lattice of binary strings where for any two strings *a* and *b*, a≤b⇔a∧b=a. (**Right**): The two shaded regions correspond to the decomposition of the 3-point interaction into two 2-point interactions.

**Figure 3 entropy-25-00648-f003:**
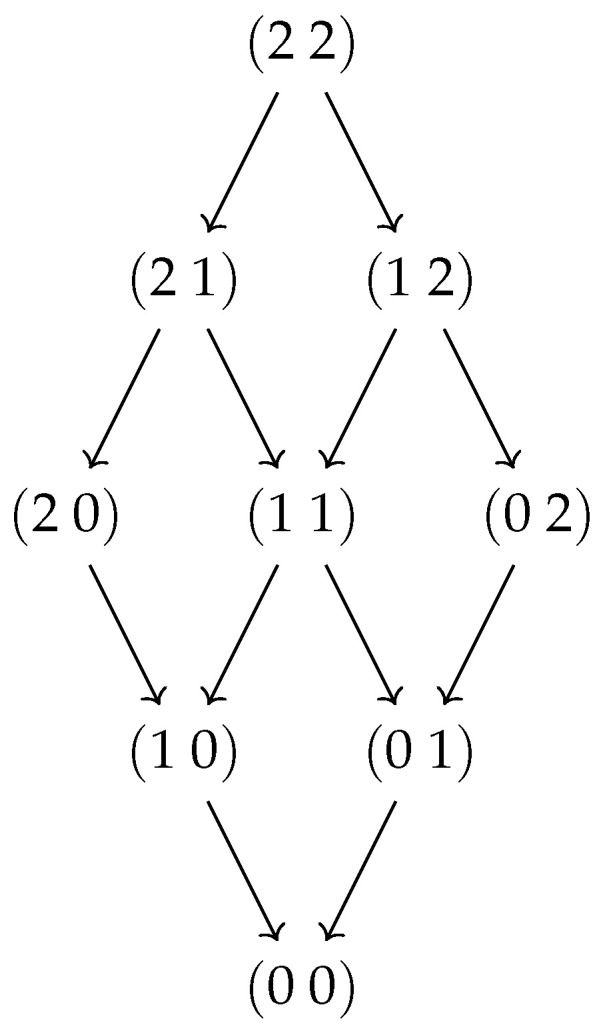
The lattice of two variables that can take three values, ordered by a≤b⇔∀i$:a_{i}\leq b_{i}$.

**Table 1 entropy-25-00648-t001:** The 3-point interactions for all two-input logic gates at equal noise level are related through $I=4\log\frac{p}{\epsilon }$ and degenerate in AND∼NOR and OR∼NAND.

G	$I_{\mathrm{ABC}}^{G}$
XNOR	*I*
XOR	−I
AND	$\frac{1}{2}I$
OR	$-\frac{1}{2}I$
NAND	$-\frac{1}{2}I$
NOR	$\frac{1}{2}I$

**Table 2 entropy-25-00648-t002:** The marginal entropies of variables in a logic gate are degenerate in XOR∼XNOR and AND∼OR∼NAND∼NOR.

G	$\begin{array}{c} H(A)\\ =H(B) \end{array}$	H(C)	H(A,B)	$\begin{array}{c} H(A,C)\\ =H(B,C) \end{array}$	H(A,B,C)
XNOR	1	1	2	2	2
XOR	1	1	2	2	2
AND	1	$\log\frac{3^{3/4}}{4}$	2	$\frac{3}{2}$	2
OR	1	$\log\frac{3^{3/4}}{4}$	2	$\frac{3}{2}$	2
NAND	1	$\log\frac{3^{3/4}}{4}$	2	$\frac{3}{2}$	2
NOR	1	$\log\frac{3^{3/4}}{4}$	2	$\frac{3}{2}$	2

**Table 3 entropy-25-00648-t003:** While the interactions leave certain gates indistinguishable, the dual *J*-interactions of the inputs are unique to each gate. The reported decimal values are rounded to three digits; as before, $I=4\log\frac{p}{\epsilon }$.

G	${\mathrm{MI}}_{\mathrm{ABC}}$	${\mathrm{MI}}_{\mathrm{BC}}$	${\mathrm{MI}}_{A}^{*}$	$I_{\mathrm{ABC}}^{G}$	$I_{\mathrm{AB}}^{G}$	$I_{\mathrm{BC}}^{G}$	$I_{A}^{*G}$	$I_{C}^{*G}$	$J_{A}^{*G}$	$J_{C}^{*G}$	${\bar{J}}^{*G}$
XNOR	−1	0	−1	*I*	$-\frac{1}{2}I$	$-\frac{1}{2}I$	$\frac{1}{2}I$	$\frac{1}{2}I$	$\frac{3}{2}I$	$\frac{3}{2}I$	$\frac{27}{8}I^{3}$
XOR	−1	0	−1	−I	$\frac{1}{2}I$	$\frac{1}{2}I$	$-\frac{1}{2}I$	$-\frac{1}{2}I$	$-\frac{3}{2}I$	$-\frac{3}{2}I$	$-\frac{27}{8}I^{3}$
AND	−0.189	0.311	$-\frac{1}{2}$	$\frac{1}{2}I$	$-\frac{1}{4}I$	0	$\frac{1}{2}I$	$\frac{1}{4}I$	$\frac{1}{2}I$	$\frac{3}{4}I$	$\frac{3}{16}I^{3}$
OR	−0.189	0.311	$-\frac{1}{2}$	$-\frac{1}{2}I$	$\frac{1}{4}I$	$\frac{1}{2}I$	0	$-\frac{1}{4}I$	−I	$-\frac{3}{4}I$	$-\frac{3}{4}I^{3}$
NAND	−0.189	0.311	$-\frac{1}{2}$	$-\frac{1}{2}I$	$\frac{1}{4}I$	0	$-\frac{1}{2}I$	$-\frac{1}{4}I$	$-\frac{1}{2}I$	$-\frac{3}{4}I$	$-\frac{3}{16}I^{3}$
NOR	−0.189	0.311	$-\frac{1}{2}$	$\frac{1}{2}I$	$-\frac{1}{4}I$	$-\frac{1}{2}I$	0	$\frac{1}{4}I$	*I*	$\frac{3}{4}I$	$\frac{3}{4}I^{3}$

**Table 4 entropy-25-00648-t004:** The joint probability of the dyadic and triadic distributions [47]. All other states have a probability of zero.

Dyadic			
**X**	**Y**	**Z**	**P**
0	0	0	1 / 8
0	2	1	1 / 8
1	0	2	1 / 8
1	2	3	1 / 8
2	1	0	1 / 8
2	3	1	1 / 8
3	1	2	1 / 8
3	3	3	1 / 8
**Triadic**			
**X**	**Y**	**Z**	**P**
0	0	0	1 / 8
1	1	1	1 / 8
0	2	2	1 / 8
1	3	3	1 / 8
2	0	2	1 / 8
3	1	3	1 / 8
2	2	0	1 / 8
3	3	1	1 / 8

## Data Availability

No new data were created or analyzed in this study. Data sharing is not applicable to this article.

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
