# Peer review of "Higher-Order Interactions and Their Duals Reveal Synergy and Logical Dependence beyond Shannon-Information"

_entropy, 2023, doi:10.3390/e25040648_

Round 1

Reviewer 1 Report

The author connects the method to capture higher-order interactions (referred to as MFIs) in Ref.[32] to information theoretic quantities including a higher-order generalization of mutual information using the technique of Mobius inversion on lattices. Considering the dual of lattices, the notion of dual interactions called outeractions is introduced. The ability of MFIs and their duals to separate different small logical gates and different small dynamical networks is investigated theoretically and numerically, respectively, and compared with that of the quantities based on the Shannon entropy. 

The paper seems to be well-written and scientifically sound. It elucidates recent results on the quantification of higher-order interactions in complex dynamical systems in the literature from a simple unified perspective. Thus, the publication of the paper will contribute to the relevant scientific community. However, the following minor comments should be considered before publication. 

(1) Sec.2:

The definition of the Boolean derivative should be given in Sec.2 rather than in the proof of Theorem 1. 

(2) Page 8, Eq.(35):

The symbol `$\hat{1}_S$' should be explained in the main text, although it appears in Fig.1 and one can guess what it means. 

(3) Pages 12--13, two diagrams between Line 189 and Line 194:

Since these two diagrams represent how each element moves from one vector space to another, the arrows should be expressed like `$\mapsto$'. 

(4) Page 14, Line 213:

`2^3=6' should be corrected. 

Reviewer 2 Report

This paper introduces a method to relate conventional information-theory (IT) quantities and the so-called model-free interaction (MFI) that was originally introduced to infer the interactions of Ising models from observational data. It unifies these seemingly disparate measures by leveraging the Boolean algebra lattice and expressing the various quantities in terms of Mobius functions defined over the lattice both in its conventional and inverted forms. One of its main results is that MFI is better able to distinguish various logic gates and causal structures compared to conventional IT measures. This is a well-written paper that uses relatively simple language that renders the material mostly accessible. While the thrust of the paper is mostly clear, it might leave one wondering about the deep reason why MFI has an edge over IT. The author briefly attributes it to the point-wise nature of MFI compared to the averaging nature of IT and the connection to the RBM, but they are not satisfying. I would encourage the author to explore it further or at least speculate more elaborately in the Discussion. From my perspective, it seems that it's the causal (do-calculus-like) nature of MFI that gives it the edge over MI which is correlational in nature. In Ref [32] where MFI was originally introduced, it's cast as the "average treatment effect", which the author mentions in the Introduction, which highlights its causal nature, though neither Ref [32] nor the author of this paper delve deeper into it. Of course, I'm not inviting philosophical discussions of causality here, but I'm specifically wondering if the "interventional" nature of MFI gives it the edge over IT. Here's one way to think about it. We know that logic gates can be expressed using Taylor expansion involving derivatives similar in form to those used in the paper, except the numerator involves only the output and not the joint distribution of all variables; see https://www.biorxiv.org/content/10.1101/2021.12.22.473903v3 or various other similar decompositions described in Ryan O'Donnell's book "Analysis of Boolean functions" (also, the Mobius inversion used in this paper is reminiscent of Fourier decomposition described in this book). My view is that if these causal derivatives form the foundations of logic gates then interaction measures that employ those same terms must have an edge. If the author agrees with this view or other perspectives that might help the reader to understand why MFIs have a better resolving power then this paper would feel more complete and satisfying. I would also like the author to address the following miscellaneous concerns, mostly for clarification purposes.

i. Ref [32] that this paper adopts the MFI definition from uses the log operator only to equate the "multiplicative" MFI to the "additive" MFI. Perhaps it would be useful to make it clear that the author is referring to the additive MFI proposed in Ref [32] and that, among other benefits, the log transformation helps distinguish positive and negative interactions by assigning appropriately signed values.

ii. Eq.6 only makes sense when conceived as Boolean calculus and finite differences, since the log would disappear on the right-hand side in the continuous formulation. Making this clear would help the reader.

iii. The uninitiated might find it helpful to understand the reason behind the convention of choosing 1^hat to represent the top of the lattice with 1^hat and 0^hat to the bottom and what those labels represent. This is especially since 0 and 1 are duals in Boolean algebra and therefore one does not have supremacy over the other.

iv. I can see why the author felt forced to choose the term "outer-action" to refer to dual MFI and as a result "in-action" to refer to the conventional MFI. Although, to some, they might seem like misnomers especially since "in" and "inter" have different meanings (especially from a systems perspective). To that end, I would encourage the author to simply use the term "dual interaction" to refer to the dual MFI and "interaction" to refer to the original. Or something along those lines that are not misleading. 

v. Even though Eq. 49 makes sense in analytical terms, how does one "leave out" a variable from a dataset in practice? Take the average of the rest for all possible values of the left-out variable?

vi. Unlike MI*, I* does not seem to "leave out" X since it's simply set to (conditioned on) 0 (line 182). If this is correct, it would help to explicitly state it to avoid misinterpretations. It would also help to explain the intuition behind why a variable is indeed "left out" in MI* but conditioned on a particular value for I*.

vii. It's interesting that even though I* is a more nuanced version of MFI (Eq.54 makes that explicit) it has less resolving power, at least for I*_A as evidenced in Table.3. Does I*_C also have the same resolution? In a similar vein, how does MI* fair in comparison to MI? Expanding Table.4 with these quantities would be helpful in that regard.

viii. In fact, if the intuition that MFI has better resolution than IT because of its "interventional" aspect, the dual quantities should be even better because of the additional intervention due to "leaving out" or additional conditioning. But it doesn't really seem to be the case. Some speculation as to what might be going on here would help hone the reader's intuition.

ix. How is the following possible (line 198)? "That means that there are no systems with zero pairwise mutual information, yet positive higher-order information. This is not true for the interactions: each interaction, at any order, is free to vary independently, and each combination of interactions defines a unique probability distribution" It's counterintuitive since it makes sense to assume that every joint probability distribution would imply constraints within their projections for subsets of variables unless it's a uniform distribution.

x. Please clarify what the dynamics are for the fork and the chain (line 244)? Just copy rules?

xi. In Fig.4, why do some MFIs not have a 3-point quantity (circles)? Is it that they are zero because of the non-multiplicative interactions? If yes, please make that clear. Also, how is partial correlation computed and why is it suddenly introduced?

xii. How's the expression J*_A in line 229 to be interpreted since it follows a different sign convention that doesn't seem to follow from any analytical treatment? Also, it seems to be measuring the effect of including A, not C. Why is it that J*_A has more resolving power compared to J*_C?

xiii. In line 262: MI doesn't suffer from this drawback of inferring a dependency among the parent nodes in a collider, a crucial difference that must be noted in the discussion.

xiv. Typos: The conditional |z=0 is missing in the term following '-' in the expression in para 147; 2^3=6 in line 214.

Round 2

Reviewer 1 Report

The author integrated all of the reviewer's comments into the revised manuscript. The reviewer now recommends accepting the manuscript in its present form. 

Author Response

The author thanks the reviewer for their helpful comments. 

Reviewer 2 Report

I thank the author for clarifying my queries and responding to my comments. I'm satisfied with most of the responses, except I would still love to get some additional clarification on the below. That said, I'm not requesting another round of revision, as these concerns are not central to the paper; I'll trust that the author will respond to those in the final version.

i. In response to point (9), the author explains how it's possible to define a distribution explicitly with 3-point interactions. However, it's still not obvious to me how it rules out 2-point interactions. Is it not possible that there may be constraints in the projected 2-variable distributions, and therefore non-zero 2-point MFI if one were to measure those?

ii. The reasoning outlined in the response to point (12) makes sense. There's just a typo in the denominator of eqn.2 where the term on the right side should read (p011p010), not (p101p100).

iii. In the response to point (13), the author explains that the 3-point MI also suffers from spurious values in that it's negative for colliders. The property of negativity for multivariate MI has long been debated and some have even proposed ways to meaningfully interpret it. Of course, there are even non-negative decompositions of multivariate MI that get around this issue. Therefore, just because the MI is negative doesn't mean that it's nonsensical or paradoxical. To me, what's more of a concern is whether a measure infers any interaction where there isn't one. In the case of a 3-point interaction in a logic gate, one would expect a non-zero value since there is an interaction among them. Therefore, I'm not entirely convinced by the author's response here.

Author Response

I thank the reviewer for their further comments, and will address these here. 
i) Taking the third derivative of the shown distribution with respect to X_i, X_j and X_k and conditioning on all other variables being zero immediately gives the 3-point interaction J_ijk. If one only takes derivatives with respect to two variabels, say X_i X_j, then the value of the interaction I_ij is proportional to J_ijk X_k, which vanishes upon conditioning X_k to zero. Therefore, all pairwise and linear interactions vanish, and only 3-point interactions remain. To emphasise that it is the conditioning that achieves this, I've added a sentence
"This distribution has 3-point interactions with strength $J_{ijk}$ for triplets $\{X_i, X_j, X_k\}$, but all pairwise interactions among $\{X_i, X_j\}$ vanish upon conditioning on $X_k=0$."

ii) While it is true that simply rearranging the terms would yield (p011p010), not (p101p100), here I've used the property of logic gates mentioned in the preceding sentence, namely that pijk=pjik. This results in p011=p101 and p010=p100, which then reproduces the denominator of eqn. 2. 

iii) I agree that a negative MI is not nonsensical. However, the 3-point MI is, in this case, exactly equal to the conditional pairwise MI of the 'inputs', so a direct reflection of Berkson's paradox. While hard to interpret in general, the simple case of a collider with known dynamics studied here actually reveals a very real informational effect. To steal the example from Wikipedia:
A car's engine can fail to start due to either a dead battery or a blocked fuel pump. Ordinarily, we assume that battery death and fuel pump blockage are independent events, MI(blocked fuel;dead battery)=0. But knowing that the car fails to start, if an inspection shows the battery to be in good health, we can conclude that the fuel pump must be blocked. Therefore MI(blocked fuel;dead battery|engine fails)>0, and the result is negative interaction information. (source :https://en.wikipedia.org/wiki/Interaction_information)
To hopefully make this more clear, I have changed "[...] a spurious effect already found [...]" to "[...] a phenomenon already found [...]", to emphasise that I don't mean that the effect is nonsensical.